# Risk prediction of all-cause mortality in hospitalized patients with severe acute pancreatitis by serum urea nitrogen/albumin ratio

**Huisi Qiu[1,2]☮, Yongshuai Fu[1,2]☮, Ziwen Lv[2], Xingyu Zhu[3], Junhui Gu[3], Zhuo Gao[3], Dong Liu[3]***

**1** Department of Qingyuan People,s Hospital, Guangzhuo, China, **2** Department of Guangzhou Medical University, Beijing, China, **3** Department of Nephrology, Air Force Medical Center, PLA, Beijing, China

☮ These authors contributed equally to this work and should be considered co-first authors.
* 2758206850@qq.com

## Abstract

### Background

Classification of risk levels in patients with acute pancreatitis remains a difficult task. Although some biomarkers have emerged to predict the prognosis of patients with acute pancreatitis, they have not been widely used in clinical practice for several reasons. This study aimed to investigate the correlation between the serum urea nitrogen-to-albumin ratio (BAR) and the mortality risk in ICU patients with acute pancreatitis.

### Method

This was a retrospective study. Data were collected from the Medical Information Mart for Intensive Care IV (MIMIC IV 2.0). The primary outcome was in-hospital mortality in the patients with severe acute pancreatitis. In this study, multivariate logistic proportional regression analysis and restricted cubic spline regression were used to evaluate the association between serum urea nitrogen/albumin ratio and in-hospital mortality in patients with severe acute pancreatitis. In the sensitivity analysis, the Boruta and random forest algorithms were used to determine the feature importance of the serum urea nitrogen to albumin ratio, and subgroup analysis was used to explore the robustness of the results.

### Result

726 patients were included in this study. The in-hospital mortality rate was 11.85%. Multivariate logistic regression analysis revealed that BAR was independently associated with an increased risk of in-hospital mortality (odds ratio [HR], 1.081 [95% confidence interval [CI], 1.062–1.101]; P < 0.001). The restricted cubic spline regression

**Data availability statement:** The raw data used in this study have been uploaded to the Figshare database and can be accessed via the following link: doi:10.6084/m9.figshare.28560035The dataset includes the raw patient data/data extraction codes and the codes used for statistical analyses. We also provide a data dictionary detailing the definition and units of each variable. These data are distributed under the Creative Commons Attribution 4.0 International (CC BY 4.0) licence, which permits free use and sharing, provided the original source is acknowledged.

**Funding:** This work was supported by grants from the National Military Standards Program (BKJ20B047), Air Force Medical Center Science and Technology Boosting Program (2022ZTYB46), and Air Force Medical Center Clinic Study Program (2021LC006).

**Competing interests:** The authors have declared that no competing interests exist.

model revealed that the risk of in-hospital mortality increased linearly with increasing BAR. Sensitivity analysis indicated that the results were stable. Additionally, the machine learning results suggested that the serum urea nitrogen to albumin ratio is an important feature for the outcomes of severe acute pancreatitis.

## Conclusion

A high serum urea nitrogen to albumin ratio was significantly associated with in-hospital mortality in patients with severe acute pancreatitis.

---

## 1. Introduction

Acute pancreatitis (AP) is a complex, unpredictable, and potentially fatal condition that is the leading cause of hospitalization in patients with gastrointestinal disorders [1,2]. The global incidence and mortality of acute pancreatitis have been increasing in recent years [3,4]. In addition, approximately 20% of patients with acute pancreatitis progress to moderate or severe acute pancreatitis, with an overall mortality rate of 20% to 40% [5,6]. Further identification of risk factors associated with acute pancreatitis outcomes is critical to enable early prevention strategies to reduce the incidence of adverse outcomes [7]. At present, there are many studies on indicators that reflect the severity of acute pancreatitis, such as serum cholinesterase, CRP, PCT, anion gap, arterial blood lactic acid, and other biochemical indicators [8–12], but these lack the specificity to distinguish the disease. The current acute pancreatitis severity score requires a longer time to capture the full range of indicators, potentially resulting in missed opportunities for early preventive measures [13–15]. Therefore, a simple, economical, and highly sensitive index is required to predict the outcomes. Serum urea nitrogen is a biochemical index reflecting kidney function and nutritional status [16,17], and severe acute pancreatitis is often complicated by acute kidney injury [18,19]. Several studies have reported that low albumin levels are associated with the severity and prognosis of infection [20–23]. As single indicators, serum urea nitrogen and albumin are susceptible to pathological factors and lack specificity. Recently, it has been reported that the ratio of serum urea nitrogen to albumin is associated with the prognosis of different types of diseases [24–27]. However, there is a lack of studies investigating the relationship between BAR and severe acute pancreatitis mortality. This study aimed to assess whether the BAR can be used as a predictor of mortality risk in patients with severe acute pancreatitis.

## 2. Materials and methods

### 2.1 Data source

The data in this study were obtained from MIMIC-IV (2.0), a large publicly available database of emergency physicians, intensivists, computer science specialists, and others from Beth Israel Deaconess Medical Center, MIT, Oxford University, and Massachusetts General Hospital (MGH). The Institutional Review Board of Beth Israel Deaconess Medical Center, responsible for reviewing the collection of patient

information and the creation of research resources, granted approval for waiving informed consent and endorsed the data-sharing initiative. The first author of this study (Ziwen Lv) completed the Collaborative Institutional Training Initiative (CITI) course and passed both the "Conflicts of Interest" and "Data or Specimens Only Research" (ID: 55109354). We extracted the variable of Interest for this study on June 5th, 2024. We qualified to use this database.

## 2.2 Study population

In the MIMIC-IV database, we extracted admission information for patients with acute pancreatitis according to the International Classification of Diseases Ninth Revision (ICD-9) code 577.0, and the International Classification of Diseases Tenth Revision (ICD-10) code K85-K85.92. Upon further screening, patients who meet the following criteria will be excluded: (1) Patients younger than 18 years of age at first admission; (2) patients with recurrent admissions for acute pancreatitis for whom only information on the first admission is retained. (3) patients with ICU stays of less than 24 hours; (4) patients with end-stage renal disease, HIV, or malignant neoplasms; (5) patients who had no record of urea nitrogen, serum albumin within 24 hours of admission; and (5) patients with >30% missing data for variables. Eventually, 726 patients were included in this study (Fig 1).

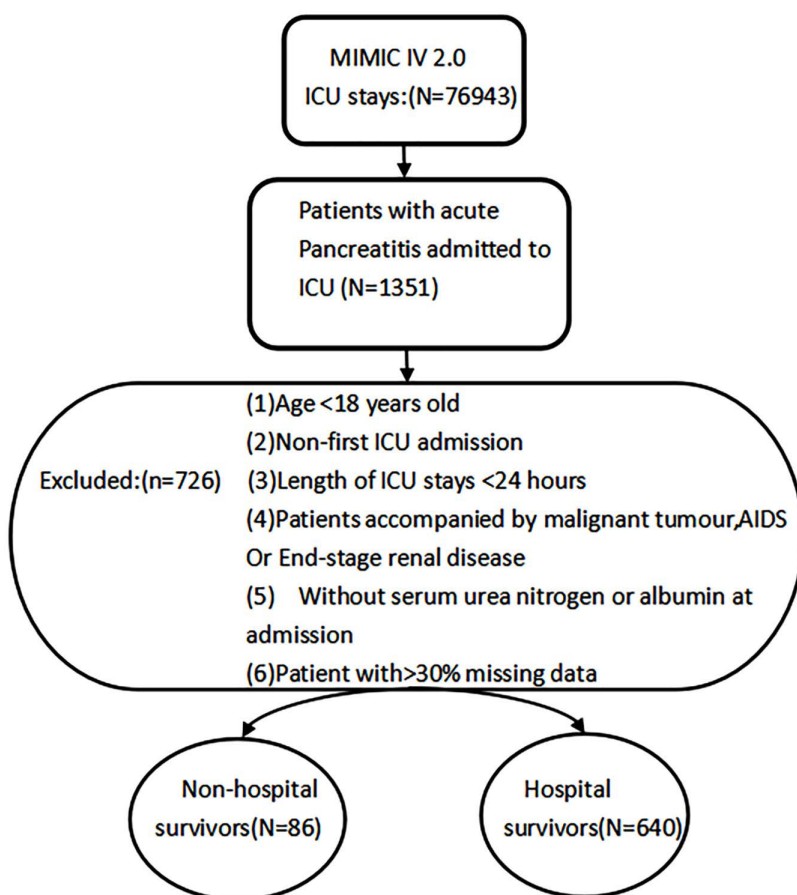

**Fig 1. Flowchart of study patients.** MIMIC, Medical Information Mart for Intensive Care, ICU, intensive care unit.

## 2.3 Data extraction

Data were extracted using Structured Query Language (SQL) run by PostgreSQL software (v13.7.1) and Navicat Premium software (version 16). Data extraction included basic population information: age, sex, ethnicity, height, and weight; clinical treatment, octreotide acetate, vasopressin, ERCP, mechanical ventilation; complications: acute kidney injury, severe sepsis, respiratory failure, heart failure, atrial fibrillation, and hepatic sclerosis; laboratory indicators: serum urea nitrogen (BUN), serum albumin, red blood cells (RBC), white blood cells (WBC), red blood cell distribution width (RDW), hematocrit, hemoglobin, total bilirubin (TBIL), blood glucose, serum creatinine, anion gap (AG), prothrombin time (PT), international normalized ratio (INR), serum potassium, serum sodium, serum total calcium, serum chloride, neutrophil ratio, lactate, alanine aminotransferase (ALT), aspartate aminotransferase (AST), bicarbonate, and platelets for the first time within 24 hours of admission and sequential organ failure (SOFA) score. Missing data is common in clinical research. The lack of data leads to the incompleteness of the original dataset, which weakens the stability and effectiveness of the research conclusions [28]. After excluding variables with more than 30% missing data, the remaining variables with missing values were processed using the multiple interpolation method. The multiple interpolation method not only provides valid estimates but also accounts for the uncertainties associated with missing data [29,30]. Assuming random data loss, we used the "mice" package in R version 4.2.3 to complete the data-filling task.

## 2.4 Feature selection

First, we used a machine-learning algorithm for feature selection. One option was to use the Boruta algorithm. The Boruta algorithm is a commonly used feature-screening algorithm that can automatically identify and select important features and has good stability. The algorithm constructs multiple random forests using datasets, calculates the importance of each feature in each random forest, and introduces shadow features to calculate the importance of the original feature and its corresponding shadow feature. If a feature is significantly more important than its corresponding shadow feature, then that feature is important [31]. We also used Shapley Additive Explanations (SHAP), a machine learning algorithm for explaining model predictions. It provides SHAP values for each feature, visualizes the importance of variables, and provides an explanation of the model's predictions [32].

## 2.5 Statistical analysis

The normality-Smirnov test of continuous variables uses continuous variables as continuous variables of the measurement error (normal distribution) and describes continuous variables of continuous variables with non-normal distribution and quantity (%). At the same time, the Kruskal-Wallis rank test was used to test the continuous variable of the continuous variable, and the non-normal distribution and value (%) were used to test the continuous variable. According to the Receiver Operating Characteristic curve (ROC), the optimal cut-off value of 10.45, and the BAR was divided into low-level and high-level groups with this cut-off value (S1 Fig). We investigated the potential nonlinear relationship between BAR levels and in-hospital mortality using a restricted cubic spline analysis. To evaluate the relationship between BAR and in-hospital mortality, a multiple logistic regression analysis was performed, and the corresponding 95% confidence interval was calculated to quantify the effect of BAR on in-hospital mortality. Model 1 included only BAR without any adjustment. Model 2 included age, sex, race, and weight for adjustment. Model 3 variable selection was adjusted according to clinical expertise and important features selected using the Boruta and SHAP algorithms (Fig 2). The variables included sequential organ failure assessment, serum albumin level, red blood cell count, white blood cell count, red blood cell distribution width, prothrombin time, alanine aminotransferase, aspartate aminotransferase, international normalized ratio, neutrophil ratio, lactate, platelets, anion gap, potassium, bicarbonate, sodium, serum creatinine, total bilirubin, vasopressin, severe sepsis, acute renal injury, and heart failure. In addition, subgroup analyses were performed to verify the association between the BAR and in-hospital mortality in each subgroup. Statistical analysis was performed using the R software (4.3.2) and SPSS Statistics (26). Statistical significance was defined as follows. $P < 0.05$, with all tests being two-tailed.

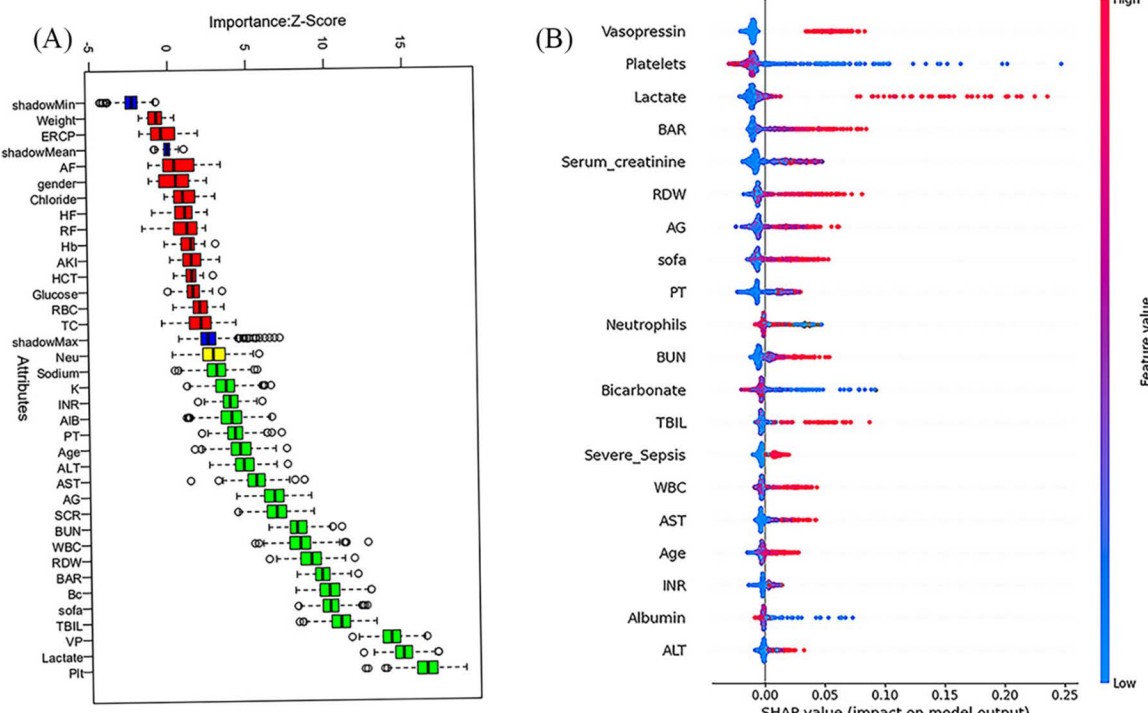

**Fig 2. Application of machine learning in feature selection.** (A) Feature selection for the relationship between various serum urea nitrogen to albumin ratio indices and in-hospital mortality was analyzed using the Boruta algorithm. The vertical axis shows the name of each variable, whereas the horizontal axis represents the Z-value of each variable. The box plot depicts the Z-value of each variable in the model calculation, with green boxes representing important variables, yellow boxes representing tentative attributes, and red boxes representing unimportant variables. (B) A Shapley Additive Explanations (SHAP) for the random forest model. Distribution of the impact of each feature on the model output. Each dot represents a patient in a row. The colors of the dots represent the feature values: red represents larger values and blue represents lower values. SOFA: sequential organ failure assessment, RBC: red blood cell, WBC: white blood cell, RDW: red blood cell distribution width, Hb: hemoglobin, TBIL: total bilirubin, INR: international normalized ratio, PT: prothrombin time, ALT: alanine aminotransferase, AST: aspartate aminotransferase, BUN: Blood urea nitrogen; AG: anion gap, TC: total calcium, Neu: neutrophil, HCT: hematocrit, SCR: Serum creatinine, K: serum potassium, Alb: albumin, Bc: bicarbonate, Plt: platelets, BAR: serum urea nitrogen to albumin ratio, RF: respiratory failure, HF: heart failure, VP: vasopressin, ERCP: endoscopic retrograde cholangiopancreatography, AF: atrial fibrillation, AKI: acute injury kidney.

## 2.6 Institutional Review Board statement

The data used in this study were collected from the MIMIC IV (v2.0) database. This was calculated by the Massachusetts Institute of Technology, Physiology Laboratory (https://physionet.org/content/mimiciv/2.0) for the development and management of public access to the database. To protect the privacy of patients, all personal information is de-identified, with a random code replacing the patient's identity, so we do not need the patient's informed consent or ethical approval. All procedures involving human participants were performed according to the ethical standards of the institutional and national research committee and with the 1964 Helsinki Declaration and its later amendments or comparable ethical standards.

## 3. Result

### 3.1 Baseline characteristics of patients

A total of 726 patients with acute were admitted to the ICU were included according to the inclusion and exclusion criteria. The in-hospital mortality rate was 11.85%. Table 1 compares the BAR ≥ 10.45 high-level group with the BAR < 10.45,

low-level group, and it was found that the patients in the high-level group were older; platelets and serum albumin were lower, while lactate, anion gap, serum urea nitrogen, alanine aminotransferase, aspartate aminotransferase, creatinine, and blood glucose levels were higher. The prevalence of acute kidney injury, severe sepsis, and heart failure was higher and more patients were treated with octreotide and vasopressin. The 1-year and in-hospital mortality rates were higher. Comparison between the hospital survivor and non-survivor groups showed that non-survivors presented with more comorbidities and higher SOFA scores. The results showed that age, race, lactate level, platelet count, serum sodium, potassium, alanine aminotransferase, aspartate aminotransferase, red blood cell distribution width, international normalized ratio, serum chloride, serum albumin, anion gap, bicarbonate, serum urea nitrogen, total bilirubin, prothrombin time, BAR, and serum creatinine levels were correlated with in-hospital mortality. Vasopressin, octreotide, and mechanical ventilation were associated with in-hospital mortality. The prevalence of death among patients with severe sepsis and acute kidney injury prevalence significantly increased (Supplementary Table 1)

### 3.2 BAR and in-hospital mortality

Logistic regression analysis showed that BAR, as a continuous variable, was independently associated with the risk of hospital death in patients with severe acute pancreatitis (HR 1.081 [95% CI 1.02–1.101]; P < 0.001). Further confirmation was made after adjusting for Models 2 and 3. The in-hospital mortality HR of the high-level BAR group was 1.050 ([95% CI: 1.026–1.074], after adjusting for the model, and was still significantly associated with hospital mortality. However, the low BAR level group had no significant correlation with the risk of death in the hospital after adjustment for Models 2 and 3 (P > 0.05) (Table 2). In addition, the restricted cubic spline regression model showed that the risk of in-hospital mortality increased linearly with increasing BAR (Fig 3).

### 3.3 Subgroup analysis

To confirm the association between BAR and in-hospital mortality, stratified analyses were performed according to age, sex, SOFA score, acute kidney injury, and severe sepsis (Fig 4). BAR was significantly associated with in-hospital mortality in men (HR = 5.01, 95% CI 2.09–12), age ≤ 70 years (HR = 4.33, 95% CI 1.9–9.87), SOFA score ≤ 7 (HR = 3.92, 95% CI 1.6–9.63), patients without acute kidney injury (HR = 3.37, 95% CI 1.16–9.77), and patients without severe sepsis (HR = 3.29, 95% CI 1.39–7.78) (P < 0.05). Stratified analysis consistently showed similar associations between the BAR and in-hospital mortality in most subgroups.

## 4. Discussion

The purpose of this study was to verify the ability of the first-admission laboratory indicator BAR to predict all-cause mortality in ICU patients with acute pancreatitis. In this study, we observed a significant association between high BAR levels and increased in-hospital mortality in patients with severe acute pancreatitis. After further adjustment for confounding factors, the predictive power of high BAR levels for in-hospital mortality in patients with severe acute pancreatitis remained strong. We used BAR as a continuous variable for ROC curve analysis, and the results showed that the AUC of BAR was 80.1% (95% CI: 75.4%−84.8%), which had significant predictive results. Our results are helpful in clinical practice for the early intervention of acute pancreatitis patients with a high risk of death to reduce in-hospital mortality.

Recent studies have shown that BAR can be used as a potential marker for cardiovascular diseases, lung diseases, sepsis, kidney injury, and other diseases [33–37]. Many clinical studies have confirmed that high levels of BAR are significantly associated with poor prognosis in critically ill patients with different disease types. In 2021, Ryu et al. found that a high BAR level was an independent risk factor for 28-day mortality in patients with aspiration pneumonia [37]. In 2021, Zou et al. observed that BAR had a strong potential predictive ability for 30-day mortality in patients with Escherichia coli bacteremia [38]. Collectively, these studies suggest a potential association between BAR and clinical outcomes in

**Table 1. Characteristics and outcomes of participants categorized by BAR.**

| | BAR < 10.45(414) | BAR ≥ 10.45(312) | P-value |
|---|---|---|---|
| Age | 51.3±16.9 | 60.3±16.1 | **<0.001** |
| Weight/Kg | 85.6±24.4 | 86.97±23.3 | 0.086 |
| Height/cm | 168.0(163.0,178.0) | 170.0(163.0,178.0) | 0.517 |
| RBC(m/uL) | 3.9(3.5,4.6) | 4.0(3.5,4.5) | 0.455 |
| Lactate(mmol/L) | 2.1(1.5,3.4) | 3.7(2.4,6.2) | **<0.001** |
| WBC(K/uL) | 19.8(15.0,25.7) | 24.3(17.6,33.2) | 0.484 |
| Platelets(K/uL) | 460.5(311.0,658.0) | 427.0(247.0,578.0) | 0.474 |
| Sodium(mEq/L) | 143.0(141.0,147.0) | 148.0(144.0,152.0) | **<0.001** |
| Total Calcium(mg/dL) | 9.0(5.6,9.5) | 9.2(8.8,9.9) | **<0.001** |
| Hemoglobin(g/L) | 12.1(10.7,14.2) | 12.2(10.7,13.6) | 0.478 |
| RDW(%) | 15.3(14.3,17.4) | 17.8(16.0,20.3) | **<0.001** |
| ALT(IU/L) | 64.5(35.0,137.0) | 121.0(43.0,306.0) | **0.020** |
| Potassium(mEq/L) | 4.8(4.5,5.2) | 5.4(4.9,6.1) | **<0.001** |
| Neutrophils | 86.0(78.9,90.0) | 89.0(84.0,92.0) | **<0.001** |
| INR | 1.5(1.3,1.9) | 2.0(1.5,2.7) | **<0.001** |
| HCT (%) | 36.2(33.0,41.7) | 37.0(32.8,41.9) | 0.898 |
| Chloride (mEq/L) | 110.0(106.0,114.0) | 113.0(110.5,118.0) | **<0.001** |
| BUN (mg/dL) | 20.0(13.0,26.0) | 71.0(49.0,102.0) | **<0.001** |
| Glucose(mmol/L) | 190.5(155.0,260.0) | 279.0(206.0,441.0) | **<0.001** |
| Albumin(g/dL) | 3.2(2.9,3.7) | 3.1(2.7,3.6) | **0.001** |
| Serum creatinine(mg/dL) | 1.0(0.8,1.3) | 3.1(1.9,5.0) | **<0.001** |
| AG (mEq/L) | 18.0(16.0,20.0) | 22.00(18.0,27.0) | **<0.001** |
| Bicarbonate(mEq/L) | 30.0(28.0,33.0) | 30.0(27.0,33.0) | 0.467 |
| TBIL (mg/dL) | 1.5(0.8,3.6) | 2.8(1.1,7.0) | **<0.001** |
| PT(s) | 16.4(14.3,21.1) | 21.7(16.5,29.2) | **<0.001** |
| AST(IU/L) | 86.0(50.0,198.0) | 182.0(76.0,502.0) | **<0.001** |
| sofa | 5.0(3.0,8.0) | 10.0(7.0,13.0) | **<0.001** |
| Gender male | 225(54.3%) | 197(63.1%) | **0.017** |
| Death during hospitalization | 13(3.1%) | 73(23.4%) | **<0.001** |
| 1-year death | 28(6.8%) | 44(14.1%) | **0.001** |
| Octreotide Acetate | 33(8.0%) | 50(16.0%) | **<0.001** |
| Vasopressin | 22(5.3%) | 102(32.8%) | **<0.001** |
| ERCP | 15(3.6%) | 10(3.2%) | 0.760 |
| Mechanical ventilation | 178(43.0%) | 413(56.9%) | **<0.001** |
| RF | 113(27.3%) | 180(57.7%) | **<0.001** |
| HF | 59(14.3%) | 70(22.4) | **0.004** |
| Hepatic sclerosis | 29(7.0%) | 44(14.1%) | **0.002** |
| Severe Sepsis | 65(15.7%) | 145(46.5%) | **<0.001** |
| AKI | 79(23.7%) | 255 (81.7%) | **<0.001** |
| AF | 55 (13.3%) | 91 (29.1%) | **<0.001** |

SOFA:sequential organ Failure assessment; RBC:red blood cell; WBC:white blood Cell; RDW: red blood cell distribution width; TBIL: total bilirubin; INR: international normalized ratio; PT:prothrombin time; ALT:alanine aminotransferase; AST:aspartate aminotransferase; BUN: blood urea nitrogen; AG: anion gap; BAR: urea nitrogen/albumin ratio; RF: respiratory failure; ERCP:endoscopic retrograde cholangiopancreatography; HF:heart failure;AF:atrial fibrillation; AKI: acute injury kidney; P – value less than 0.05 is expressed in bold.

**Table 2. The association between BAR groups and in-hospital mortality.**

| Variables | HR | 95%CI | P-value |
|---|---|---|---|
| Model I | | | |
| BAR | 1.081 | 1.062-1.101 | <0.0001 |
| BAR<10.45 | 1.298 | 1.005-1.676 | 0.046 |
| BAR≥10.45 | 1.050 | 1.026-1.074 | <0.0001 |
| Model II | | | |
| BAR | 1.080 | 1.060-1.100 | <0.0001 |
| BAR<10.45 | 1.195 | 0.903-1.580 | 0.213 |
| BAR≥10.45 | 1.052 | 1.027-1.078 | <0.0001 |
| Model III | | | |
| BAR | 1.068 | 1.020-1.111 | 0.001 |
| BAR<10.45 | 1.134 | 0.708-1.815 | 0.601 |
| BAR≥10.45 | 1.048 | 1.003-1.096 | 0.037 |

BAR: urea nitrogen/Albumin Ratio. CI, confidence interval.

Model I wasn't adjusted.

Model II was adjusted for age, gender, ethnicity, and weight.

Model III was adjusted for sequential organ failure score, red blood cell distribution, width, red blood cells, white blood cells, prothrombin time, alanine aminotransferase aspartate aminotransferase, international normalized ratio, neutrophil ratio, lactate, platelets, anion gap, potassium, bicarbonate, sodium, serum creatinine, total bilirubin, vasopressin, acute renal injury, severe sepsis, and heart failure.

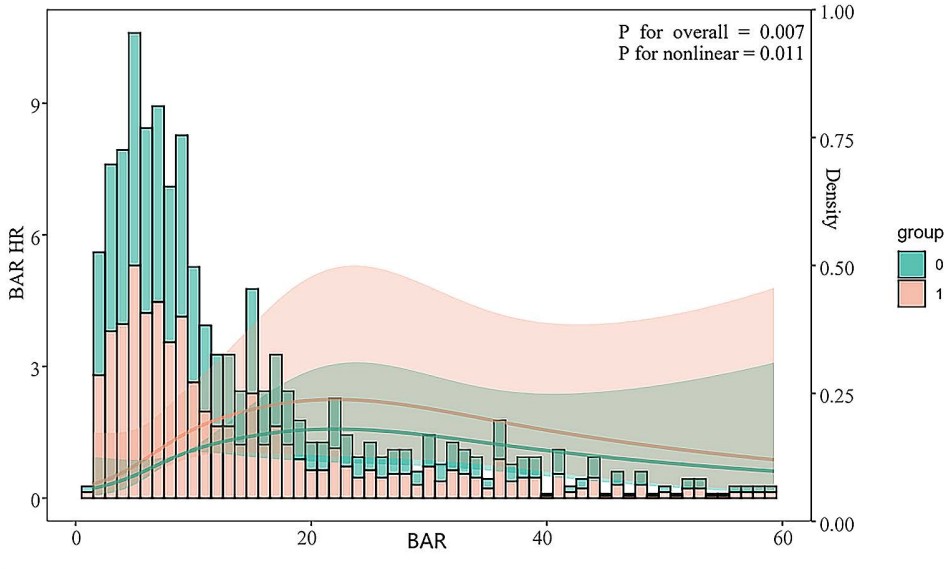

**Fig 3. Restricted cubic spline relationship between serum urea nitrogen to albumin ratio and in-hospital mortality risk.** 1 in Group grouping in Figures represents BAR greater than or equal to 10.45, and 0 represents less than 10.45. The sizes of BAR are shown separately on the x-axis. The distribution of BAR density in the study population is represented by the histogram.

critically ill and infection-related patients. The results of a large 2022 study that included 13,464 patients with sepsis found that patients in the highest quartile of BAR had an increased risk of sepsis-related death compared to those in the lowest quartile of BAR. Moreover, when BAR was used as a continuous variable, the incidence of hospital sepsis-related deaths increased by 8% for every 5-unit increase in BAR [36].

| Variable | N | HR(95%CI) | Death.during.hospitalization | P.value |
|----------|---|-----------|------------------------------|---------|
| overall | 726 | 3.22(1.79-5.9) | | <0.001 |
| Age | | | | |
| >70 | 184 | 1.12(0.45-2.78) | | 0.801 |
| <=70 | 541 | 4.33(1.9-9.87) | | <0.001 |
| gender | | | | |
| female | 304 | 5.01(2.09-12) | | <0.001 |
| male | 422 | 2.31(0.99-5.35) | | 0.052 |
| sofa | | | | |
| >7 | 249 | 1.97(0.84-4.6) | | 0.119 |
| <=7 | 477 | 3.92(1.6-9.63) | | 0.003 |
| AKI | | | | |
| No | 392 | 3.37(1.16-9.77) | | 0.025 |
| Yes | 334 | 1.71(0.74-3.98) | | 0.212 |
| Severe Sepsis | | | | |
| No | 516 | 3.29(1.39-7.78) | | 0.007 |
| Yes | 210 | 1.83(0.78-4.33) | | 0.168 |

**Fig 4. Subgroup analyses for the association of BAR with in-hospital mortality, AKI: acute kidney injury, SOFA: sequential organ failure assessment.**

The results of our study suggest that high levels of BAR are associated with severe acute pancreatitis in patients. Activation of the inflammatory response and recruitment of inflammatory cells in acute pancreatitis lead to tissue damage and exacerbation of the disease [39]. The prognosis of acute pancreatitis is closely related to the severity of the inflammatory response, and BAR is a new prognostic indicator reflecting body inflammation that has attracted much attention in recent years. Our study showed that BAR was positively correlated with disease severity. BAR reflects the severity of the disease in patients with acute pancreatitis admitted to the ICU, allowing clinicians to understand the disease condition of patients at an early stage and contributing to the clinical management of patients with severe acute pancreatitis. The biological mechanism underlying the relationship between BAR and prognosis in patients with acute pancreatitis remains unclear. When inflammation occurs, the body consumes a large amount of albumin to increase the production of anti-inflammatory substances, such as lipoxins, lysins, and protectins, to promote wound recovery [40]. Changes in BUN reflect changes in blood concentration, microcirculation, impaired kidney function, and increased protein catabolism [41–43]. Changes in serum urea nitrogen and albumin levels are closely associated with the occurrence and progression of SAP. From the baseline data, we observed significant differences in SOFA scores among patients with BAR in the different groups, suggesting a strong association between BAR and disease severity. Changes in the BAR in patients with severe acute pancreatitis can reflect the inflammatory status or severity. In the sensitivity analysis, the linear relationship between BAR and in-hospital mortality in patients with severe acute pancreatitis was consistent in the young group, female patients, SOFA score ≤7, patients without sepsis, and patients without acute kidney injury. This result may be due to high SOFA scores, advanced age, sepsis, and AKI, which have traditionally been considered adverse prognostic risk factors for severe acute pancreatitis. In addition, high SOFA scores, advanced age, AKI, and sepsis may lead to high BAR values, leading to an underestimation of the association between BAR and mortality in patients with severe acute pancreatitis. The association between BAR and in-hospital mortality in patients with severe acute pancreatitis was consistent in women, but no studies at home and abroad have reported that the severity of acute pancreatitis and the level of BAR are related to sex. The exact mechanism underlying this finding should be explored in future studies. In addition, we conducted a feature analysis using the SHAP and Brouta algorithms and evaluated the importance of BAR as a feature in the outcome prediction model. In the future, a machine learning prognostic model for patients with severe acute pancreatitis can be established by focusing on the BAR.

However, our study has its limitations. Firstly, our study was retrospective, thus precluding the explicit establishment of causality. Although we used a range of rigorous statistical methods to produce robust results. Second, our study measured the relationship between BAR and prognosis for the first time after admission, which made it impossible to assess the effect of dynamic BAR on prognosis. Third, due to the presence of missing variables in the database, un-adjusted confounders may have affected our results despite multivariate adjustment and subgroup analyses. Fourth, the study population was selected from patients between 2008 and 2019, a period during which medical advances and optimization of treatment regimens did not guarantee consistent patient treatment, which may have affected the results. Finally, prospective cohort studies are necessary to validate our findings.

## 5. Conclusion

Our results suggest that high BAR levels are strongly associated with increased in-hospital all-cause mortality in patients with severe acute pancreatitis. BAR may be used as a predictor of mortality in patients with severe acute pancreatitis, which is helpful for clinicians in stratifying risk and intervening early in patients with severe acute pancreatitis to reduce in-hospital mortality.

## Author contributions

**Conceptualization:** Ziwen Lv.

**Data curation:** Ziwen Lv, Junhui GU.

**Formal analysis:** Ziwen Lv, Xingyu Zhu.

**Funding acquisition:** Dong Liu.

**Investigation:** Xingyu Zhu, Dong Liu.

**Software:** Xingyu Zhu.

**Visualization:** Zhuo Gao.

**Writing – review & editing:** Huisi Qiu, Yongshuai Fu.

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
