## [Decision Letter · Decision Letter 0]

27 Jan 2025

Dear Dr. Lv,

Thank you for submitting your manuscript to PLOS ONE. After careful consideration, we feel that it has merit but does not fully meet PLOS ONE’s publication criteria as it currently stands. Therefore, we invite you to submit a revised version of the manuscript that addresses the points raised during the review process.

We look forward to receiving your revised manuscript.

Kind regards,

Ramya Iyadurai

Academic Editor

PLOS ONE

Journal Requirements:

“This work was supported by grants from the National Military Standards Program (BKJ20B047), Air Force Medical Center Science and Technology Boosting Program (2022ZTYB46), and Air Force Medical Center Clinic Study Program (2021LC006).”

5. We note that your Data Availability Statement is currently as follows: [All relevant data are within the manuscript and its Supporting Information files.]

6. Please amend the manuscript submission data (via Edit Submission) to include authors Dr. Junhui Gu and Dr. Zhuo Gao.

7. Please amend your authorship list in your manuscript file to include author Dr. Qi Feng.

8. Please include captions for your Supporting Information files at the end of your manuscript, and update any in-text citations to match accordingly. Please see our Supporting Information guidelines for more information: http://journals.plos.org/plosone/s/supporting-information .

Additional Editor Comments:

BAR ratio in acute pancreatitis outcomes

I would like to appreciate the authors for an interesting study, the importance of the study I believe lies in the fact that a simple and easily available blood test can be used to predict mortality in a very severe illness.

I have a few recommendations for the manuscript,

Recommendation 1 :

The results can include the numerical values, the percentages and other interesting statistics baseline characteristics of the study group, the mean age of the study group, the gender distribution, significant comorbidities, substance use, the etiology for the pancreatitis, outcomes, complications, and other values of interest in acute pancreatitis like blood sugar levels, calcium levels, SOFA scores, this will easier for the reader to read and assimilate if included in the manuscript, instead of having to refer to the tables at each point.

In the results the authors can discuss what factors were significantly different and provide numerical values for clarity for the readers.

Recommendation 2 :

The authors can also discuss in the results on whether they identified any other factors they found of interest in the study that could be interesting for the readers and lead to future studies.

Recommendation 3:

The authors can discuss the results under models 2 and 3, especially model 3 clarify what they indicate by mentioning clinical expertise, how this was derived for scientific interest.

The authors can mention and explain the factors that were selected by the Baruta and the SHAP algorithm in the results.

It will be of interest of the authors explain the figures on the importance of Z score, the SHAP and the BAR, the BAR figure has a legend of 2 groups 0,1 but it is not clear what is 0 and what is 1, can the authors label the legends of the figure for more clarity.

The plot of HR is pictorially represented as a forest plot, but usually forest plots are usually drawn with 0 and values on either side, for significance, here the plot is represented with 1, so it may appear to the readers that values are nor significant when they actually are, can authors look into this and if they do not want to change the present representation explain the same in the results so that it will be clearer for the readers.

Recommendation 4

It will be useful for the readers if the authors can mention the subgroups chosen for analysis and the reasoning behind their choice of the subgroups for analysis in the methodology. The authors can mention in the results the values for the subgroups that are not significant as well, since in the results only the factors that were significant are mentioned.

Recommendation 5:

The authors have excluded patients with HIV, kidney disease and malignant neoplasms, I assume this is probably since the patients would be sicker and have low albumin, Can the authors clarify this in the manuscript, as it will be of value to the readers.

Recommendation 6:

Under the discussion the authors can discuss the results of their study and the similarities of baseline characteristics of their study group with other studies in pancreatitis, for the readers to identify the applicability of the study in their clinical practice.

The authors can discuss the varied values of BAR in various diseases rather than its importance in other diseases in a general way and limit it to diseases similar to pancreatitis such as illnesses with multi organ dysfunction such as sepsis.

The authors can discuss BAR and the cut off values to be used in pancreatitis and the why behind the cutoff values, for clarity for the readers.

Looking forward to resubmission after addressing the recommendations.

Reviewers' comments:

Reviewer's Responses to Questions

**Comments to the Author**

1. Is the manuscript technically sound, and do the data support the conclusions?

Reviewer #1: Yes

2. Has the statistical analysis been performed appropriately and rigorously?

Reviewer #1: Yes

3. Have the authors made all data underlying the findings in their manuscript fully available?

Reviewer #1: Yes

4. Is the manuscript presented in an intelligible fashion and written in standard English?

Reviewer #1: Yes

Reviewer #1: The study establishes a correlation between BAR and mortality, it is essential to address how this biomarker compares with existing prognostic tools used in acute pancreatitis.

The statistical methods employed are good. However providing confidence intervals for all key statistics would improve the comprehensiveness of the results.

Discussing the feasibility of implementing routine BAR testing in clinical settings, including cost-effectiveness and accessibility issues, would provide practical insights for clinicians.The reported odds ratio for BAR is significant, however, including how BAR can be integrated into current clinical practice would enhance the discussion.Overall, this study presents significant findings regarding the serum urea nitrogen-to-albumin ratio as a predictor of mortality in acute pancreatitis patients. Addressing these comments will enhance clarity, depth, and applicability of the research outcomes.

**Do you want your identity to be public for this peer review?** For information about this choice, including consent withdrawal, please see our Privacy Policy

Reviewer #1: **Yes: ** Nawahirsha

---

## [Author Response · Author response to Decision Letter 1]

11 Mar 2025

Reviewer 1

Added or modified in the revised manuscript as requested

Reviewer 2

Comment 1

Results can include numerical values, percentages, and other interesting statistical baseline characteristics for the study group, mean age of the study group, gender distribution, significant comorbidities, drug use, etiology, outcomes, complications of pancreatitis, and other values of interest in acute pancreatitis, such as blood glucose levels, calcium levels, SOFA scores, if included in the manuscript, It will be easier for the reader to read and absorb than to refer to the table at every point.

In the results, the authors can discuss which factors differ significantly and provide numerical values so that the reader can see them.

Response

Thank you very much for your suggestion,We think this is an excellent suggestion. We carefully reviewed the literature and made changes to the discussion section of the revised manuscript. Details have been added in P12-13, Line 239~264.

Comment 2

Authors can also discuss in the results whether they found other interesting factors in the study that might be interesting to the reader and lead to future research.

Response

Thank you very much for your suggestion,We think this is an excellent suggestion. We carefully reviewed the literature and made changes to the discussion section of the revised manuscript. Details have been added in P16, Line 325~330.

Comment 3

The authors could discuss the results under Models 2 and 3, specifically Model 3, by mentioning clinical expertise to clarify what they indicate and how this knowledge was derived from scientific interest.

The authors can mention and explain in the results the factors selected by the Baruta and SHAP algorithms.

Interested authors will explain the importance of the Z-score, SHP and BAR in the BAR plot, there are two groups of icons, 0, 1 but it is not clear what is 0 and what is 1, the author can annotate the plot to be clearer.

The plots of HR are represented as forest plots on the chart, but usually forest plots are usually represented by 0 and values on both sides, for the sake of meaning, the plots here are represented by 1, so the reader may feel that these values are not important, but in fact they are not important, can the author look into this issue? If they don't want to change the current presentation, they can explain the same thing in the results so that the reader will be clearer.

Response

Thank you very much for your suggestion,We think this is an excellent suggestion. We carefully reviewed the literature and made changes to the discussion section of the revised manuscript. The selection of covariates for adjustment in our model was informed by a systematic review of existing literature on mortality risk factors in acute pancreatitis, supplemented by feature importance analyses using SHAP (SHapley Additive exPlanations) and Boruta machine learning algorithms. This dual approach ensured that variables were prioritized based on both evidence from prior studies and data-driven predictive relevance.and we use R to calculate, and the HR we calculate is greater than 1, so we use starting from 1 in the forest graph.and In the early stage, we read a large number of relevant literature and summarized the relevant important variables.Following the determination of the relevance of the study variables, feature selection was a crucial step in reducing the number of features in the large dataset. A crucial method of feature selection was the Boruta algorithm, which was based on the random forest classifier method.Details have been added in P27 Line 564-566 and P11, Line 213~226.

Comment 4

It would be helpful if the authors could mention the analysis subgroup they chose and the rationale behind their methodological choice of the analysis subgroup. The authors can mention the values of unimportant subgroups in the results because only the important factors are mentioned in the results.

Response

Thank you very much for your suggestion,We think this is an excellent suggestion. We carefully reviewed the literature and made changes to the discussion section of the revised manuscript. Details have been added in P15-16, Line 306~317 and P10-11, Line 200~211.

Comment 5

The authors excluded patients with HIV, kidney disease, and malignancies, which I think may be due to the fact that the patients are more severely ill and have lower albumin. Could the author please clarify this in the manuscript as it is valuable to the reader.

Response

Thank you very much for your suggestion,We think this is an excellent suggestion. We carefully reviewed the literature and made changes to the discussion section of the revised manuscript. Details have been added in P5, Line 86~94.

Comment 6

During the discussion, authors may discuss the results of their study, as well as the similarity of the baseline characteristics of their study arm to other pancreatitis studies, so that the reader can identify the applicability of the study in clinical practice.

The authors could discuss the different values of BAR in various diseases, rather than discussing its importance in other diseases in general, and limit it to pancreatitis-like diseases, such as those with multiple organ dysfunction, such as sepsis.

The authors can discuss the BAR and cut-offs used in pancreatitis, as well as the reasons behind the cut-offs, so that it is clear to the reader.

It is expected that it will be resubmitted once the recommendations have been implemented.

Response

Thank you very much for your suggestion,We think this is an excellent suggestion. We carefully reviewed the literature and made changes to the discussion section of the revised manuscript. Details have been added in P13-14, Line 269~283 and P7, Line 134~139.

Reference Since the 40th reference in the original manuscript was required for the revision of the reviewer's comments, this reference has been placed in the 28th paragraph of the revised manuscript.

We tried our best to improve the manuscript and made some changes We appreciate for Editor and Reviewers’ warm work earnestly, and hope the correction will meet with approval. Once again, thank you very much for your comments and suggestions.

---

## [Decision Letter · Decision Letter 1]

1 Jun 2025

Risk prediction of all-cause mortality in hospitalized patients with severe acute pancreatitis by serum urea nitrogen/albumin ratio

PONE-D-24-51800R1

Dear Dr. Lv,

We’re pleased to inform you that your manuscript has been judged scientifically suitable for publication and will be formally accepted for publication once it meets all outstanding technical requirements.

Kind regards,

Ramya Iyadurai

Academic Editor

PLOS ONE

Additional Editor Comments (optional):

The font of the tables keep shifting can you please make the font uniform.

Reviewers' comments:

Reviewer's Responses to Questions

**Comments to the Author**

Reviewer #2: (No Response)

Reviewer #3: All comments have been addressed

Reviewer #4: All comments have been addressed

Reviewer #5: All comments have been addressed

2. Is the manuscript technically sound, and do the data support the conclusions?

Reviewer #2: Yes

Reviewer #3: Yes

Reviewer #4: Yes

Reviewer #5: Partly

3. Has the statistical analysis been performed appropriately and rigorously?

Reviewer #2: I Don't Know

Reviewer #3: Yes

Reviewer #4: Yes

Reviewer #5: Yes

4. Have the authors made all data underlying the findings in their manuscript fully available?

Reviewer #2: Yes

Reviewer #3: Yes

Reviewer #4: Yes

Reviewer #5: Yes

5. Is the manuscript presented in an intelligible fashion and written in standard English?

Reviewer #2: Yes

Reviewer #3: Yes

Reviewer #4: Yes

Reviewer #5: No

Reviewer #2: The author mentioned vasopressin, octreotide, and mechanical ventilation were associated with in-hospital mortality. whereas vasopressin, octreotide, and mechanical ventilation are started as a treatment when the patient's condition is worsened. So these treatment modalities would not be considered as associated factors for mortality.

so please rephrase the sentences from line 173 to 176

Reviewer #3: Comment 1,

The authors have responded to the previous review comments one by one, which has improved the overall interpretation of the research data. From the existing data, after the correction in Model 3, the BAR still shows some predictive ability for the prognosis of acute severe pancreatitis. However, it is noteworthy that during the assessment of acute severe pancreatitis, BUN and ALB as individual indicators can already indicate the severity of acute pancreatitis. The authors also mentioned the predictive efficacy of BUN and ALB in the discussion section. Based on the evidence presented in the paper, it seems that the predictive efficacy of BAR is primarily due to these individual indicators, which is reasonable as a disease stratification tool.

Comment 2,

Despite the adjustments made in the statistical model, it is undeniable that, at baseline, patients with a BAR > 10.45 have more severe conditions and higher mortality rates, which is expected. Rather than considering BAR as a single predictive factor, it more accurately reflects a state of high catabolic metabolism upon admission. However, due to the significant statistical correlation, I believe the authors' conclusions are justifiable.

Reviewer #4: The study, through rigorous methodological design and comprehensive data analysis, has confirmed the predictive value of the "Blood Urea Nitrogen to Albumin Ratio (BAR)" for in-hospital mortality risk in patients with severe acute pancreatitis (SAP). The research addresses key reviewer comments, is methodologically sound, and its conclusions are robust. It is suitable for publication.

Reviewer #5: Tables are less in number,There is no graphical representations of any of ur variables of data.IIn Data extraction heading -Why u didn't remove whole missing data,why to remove more than 30%missing variables,as it may affect the organisation of data n ur study outcomes

**Do you want your identity to be public for this peer review?** For information about this choice, including consent withdrawal, please see our Privacy Policy

Reviewer #2: **Yes: ** Jennie Santhanam

Reviewer #3: No

Reviewer #4: **Yes: ** Lei Chen

Reviewer #5: No

---

## [Editor Report · Acceptance letter]

PONE-D-24-51800R1

PLOS ONE

Dear Dr. Lv,

I'm pleased to inform you that your manuscript has been deemed suitable for publication in PLOS ONE. Congratulations! Your manuscript is now being handed over to our production team.

Kind regards,

on behalf of

Dr. Ramya Iyadurai

Academic Editor

PLOS ONE